# Identifying Elastic Constants for PPS Technical Material When Designing and Printing Parts Using FDM Technology

**DOI:** 10.3390/ma14051123

**Published:** 2021-02-27

**Authors:** Jone Retolaza, Rubén Ansola, Jose Luis Gómez, Gorka Díez

**Affiliations:** 1Bilbao Industry IX Accenture Astondo Bidea Edificio 602 Parque Tecnológico de Bizkaia, 480170 Derio, Spain; jone.retolaza@accenture.com; 2Department of Mechanical Engineering, Faculty of Engineering University of the Basque Country Alda, Urquijo s/n, 48013 Bilbao, Spain; 3Department of Materials GAIKER Technology Center Parque Tecnológico de Bizkaia Edificio 202, 48170 Zamudio, Spain; gomez@gaiker.es (J.L.G.); diez@gaiker.es (G.D.)

**Keywords:** material characterization, elastic constants, anisotropic, fused deposition modeling, finite element analysis, mechanical testing

## Abstract

This paper introduces a methodology to study the anisotropic elastic constants of technical phenylene polysulfide thermoplastic (PPS), printed using fused deposition modeling (FDM) in order to provide designers with a guide to achieve the required mechanical properties in a printed part. The properties given by the manufacturer are usually taken from injected samples and these are not the real properties for printed parts. Compared to other plastic materials, PPS offers higher mechanical and thermal resistance, lower moisture absorption, higher dimensional stability, is highly resistant to chemical attacks and environmental aging, and its fireproof performance is good. One of the main difficulties presented when calculating and designing for FDM printing is that printed parts present anisotropic behavior i.e., they do not have the same properties in different directions. Haltera-type samples were printed in the three manufacturing directions according to optimum parameters for material printing, aimed at calculating the anisotropic matrix of the material. The samples were tested in order to meet standards and values for elastic modulus, shear modulus and tensile strength were obtained, using Digital Image Correlation System to measure the deformations. An approximated transversally isotropic matrix was defined using the obtained values. The fracture was analyzed using SEM microscopy to check whether the piece was printed correctly. Finally, the obtained matrix was validated by a flexural test and a finite element simulation.

## 1. Introduction

Additive production allows pieces to be built with impossible geometries using legacy manufacturing processes. In the specific case of fused deposition modeling (FDM) printing technology, the molten filament is deposited layer by layer, thus, generating a 3D geometry with anisotropic mechanical behavior. FDM printing presents numerous challenges to overcome in order to achieve industrial implementation, such as shortage of technical materials, process control and stability, repeatability, dimensional tolerance, scalability for mass production, manufacturing speed, and modeling, with the latter being a fundamental aspect for the industry [1]. There are no accessible or reliable specifications of the mechanical properties of printed technical materials. Designing and obtaining functional parts is thus complex. Table 1 shows the mechanical properties of different commercial materials as suppliers define them [2], but that information is not frequent enough. There are many factors that affect the final quality and the properties of the printed parts during any 3D printing process. The final properties of the parts will definitely be altered by the nature, composition and microstructure of the materials, infill pattern, percentage and building direction, along with the 3D printing variables used with FDM technology.

Previous research has identified 42 influential recurring parameters associated to process, material and physical phenomena [3]:(a)As regards process: extrusion temperature, chamber temperature, printing speed, layer height, skin thickness of the printed part, the percentage and type of filling used, construction orientation regarding dimension, surface and functional capabilities related to the printed parts.(b)With respect to material: circularity and dimensional tolerance of 3D printing filaments, together with moisture content, surface roughness and internal porosity.(c)With reference to physical phenomena: room-temperature, nozzle diameter, and construction platform [3].

There is extensive research into the mechanical behavior of printed parts depending on the manufacturing parameters [4]. Ahmed et al. [5] used SEM to analyze the contour of two geometries, one with a single layer contour and the other with a 10-layer contour. They observed that the greater the number of contours, the lower the porosity and therefore, the better the mechanical performance of the part. Martí et al. [6] studied the influence of the following printing parameters in the PLA and concluded that the most influential ones in descending order are orientation, layer height and filling density. Fernandez et al. [7] concluded in their research that the traction force is directly related to the infill percentage. On the other hand, Luzain et al. [8] studied the influence of layer thickness, deposition angle and infill on flexural behavior, and concluded that layer thickness is the parameter that most influences the flexural strength of the material [9].

Zieman et al. [10] studied the relationship between the mechanical properties of ABS and the anisotropic behavior when printed by FDM. They verified that the union of layers is one of the factors that most influences elastic modulus, as better mechanical properties are achieved by modifying the printing pattern. The most appropriate manufacturing direction will be the one with joints between layers and perpendicular to the direction of the applied force, i.e., if the threads are aligned with the direction of the force, the part is more tensile resistant. Therefore, the strength of the part will improve when the orientation of the direction of the threads and the printing density is such that the threads are parallel to the direction of the stress [11].

Ridick et al. [12] went further and studied different combinations of growth direction and pattern orientation showing that they produce sections with similar orientation with very different tensile values. This is due to the way the bond between the layers works while being printed. Adhesion and properties improve when each layer is set up and reinforcement joints are formed.

Density, type of filling (infill), thickness of the skin and manufacturing orientation are the parameters that will mainly affect the mechanical properties obtained in the final product [13]. The easiest way to improve the tensile strength is to increase the number of perimeters, the thickness of the skin, and/or the infill percentage [14]. A standard infill value is 20% and a proper mechanical performance, printing time and weight of the parts is thus usually achieved. 

Regarding the type of filling, there are several alternatives available: orthogonal, triangular, honeycomb, etc. However, the most commonly filling used are orthogonal and 45-degree orientation fillings [15,16].

The majority of studies seeking to correlate manufacturing processes and final performance have usually been conducted with materials such as PLA and ABS. The number of studies decreases as we focus on high performance materials such as glass and carbon reinforced composites or PEEK, PEKK, PSU, PEI filaments. In the case of PPS, there is barely any information available, but the research by Fitzharris et al. [17] should be mentioned, which studies the warping modeling of the printed PPS parts with FDM technology.

The most recent research focuses on the study of the anisotropy of printed materials and on establishing a methodology to be able to design printed parts that meet the required technical specifications. Kibling et al. [18] determined the mechanical characteristic values of a linear orthotropic material for unidirectional FDM structures of an ABS carbon fiber reinforcement, concluding that the values of Poisson’s ratio, modulus of elasticity and shear modulus can be approached to their average to obtain the orthotropic matrix of the material. This is the hypothesis of this research, but using PPS material, rectilinear pattern and 50% infill, which are parameters that affect the properties of the material. However, there are no studies that relate anisotropic behavior of a PPS material with the FDM technology when designing printed parts. Almost all the studies so far have been conducted under the assumption that the part obtained is isotropic. The specification of the anisotropic matrix of this material and the comparison of its behavior by finite elements has not yet been performed [19]. 

The aim of this investigation is therefore twofold: on one hand, to study the behavior of PPS printing material and, on the other hand, to obtain the anisotropic matrix of the material, based on transverse isotropic material hypothesis. The validation of the proposed material model is essential as a basis of future numerical calculations from which optimal production strategies can be derived with regard to the structural component’s building-space and individual layer orientation. Finally, a validation of the results will be implemented by means of an ANSYS simulation to check the behavior of a bending beam according to the UNE-EN-ISO-178-2011 standard and contrasting these results with those obtained experimentally on printed samples [20].

## 2. Materials and Methods

In comparison with other plastic materials, PPS properties have high mechanical strength, stiffness and hardness, high thermal resistance, low moisture absorption, high dimensional stability, and are highly resistant to chemical products attacks. Additionally, it presents a very good behavior against environmental aging and fire without the need for use of flame retardant additives. Philips Petroleum Company, under the brand name of Ryton, began the industrial production of phenylene polysulfide in Texas in 1973. This is an engineering thermoplastic product with a high degree of crystallinity and high performance. PPS was initially used in the manufacture of technical parts using injection, where the required mechanical properties could withstand high temperatures in contact with chemical products. 

The present study was conducted using a PPS 3D printing filament (3NTR, Oleggio, Italy) with a 2.85 mm nominal diameter filament. Figure 1 shows the printing of a real PPS part with two different rectilinear infilling percentages. The size of the parts is 10 cm × 10 cm.

The filament section shown in Figure 2 maintains good circularity, the diameter is kept within the usual tolerance limits for 3D printing filaments and no internal porosity that may influence the mechanical properties of the printed parts is observed.

The presence of porosity in the filaments is transferred to porosity in the printed parts, thus presenting failures and mechanical weaknesses in the printed parts. Controlling the humidity in the materials to be printed and performing predrying are essential to limit process inconsistency. The processability thermal interval of the material ranges from 280 to 380 °C. The crystalline melting temperature was confirmed using thermal characterization with differential scanning calorimetry and thermogravimetry. These values mark the thermal interval of processability of the material. 

Table 2 shows the isotropic mechanical properties in the technical data sheet of the PPS material. Nevertheless, the properties of the injected parts are not equal to the properties of the 3D printed ones, as the behavior is anisotropic in 3D printing. This evidences the need for the study in this article in order to obtain useful specifications for designers. 

The PPS printing was performed using a 3ntr A2V2 printer. The 3NTR A2 printer offers a mechanical resolution of 0.015 mm and a printing accuracy of up to 0.05 mm. The following Figure 3 shows the 3D printing equipment used.

## 3. Results

### 3.1. Constitutive Relations

The constitutive equation in matrix notation for an orthotropic material in the material’s main axis system (XYZ) can be provided by the compliance matrix shown in Equations (1) and (2) The complete description requires nine independent parameters because of the tensor’s symmetry. Taking the six existing Poisson’s ratios into consideration means that only three have to be determined experimentally, since the rest of the Poisson’s ratios represent dependent quantities due to the symmetry:(1){ε}=[S]{σ}
(2)$$\left(\begin{array}{c} \varepsilon_{\mathrm{xx}}\\ \varepsilon_{\mathrm{yy}}\\ \varepsilon_{\mathrm{zz}}\\ \gamma_{\mathrm{yz}}\\ \gamma_{\mathrm{xz}}\\ \gamma_{\mathrm{xy}} \end{array}\right)=\left(\begin{array}{cccccc} \frac{1}{E_{x}} & \frac{-\upsilon_{\mathrm{xy}}}{E_{y}} & \frac{-\upsilon_{\mathrm{xz}}}{E_{z}} & 0 & 0 & 0\\ \frac{-\upsilon_{\mathrm{yx}}}{E_{x}} & \frac{1}{E_{y}} & \frac{-\upsilon_{\mathrm{yz}}}{E_{z}} & 0 & 0 & 0\\ \frac{-\upsilon_{zx}}{E_{x}} & \frac{-\upsilon_{zy}}{E_{y}} & \frac{1}{E_{z}} & 0 & 0 & 0\\ 0 & 0 & 0 & \frac{1}{G_{\mathrm{yz}}} & 0 & 0\\ 0 & 0 & 0 & 0 & \frac{1}{G_{\mathrm{xz}}} & 0\\ 0 & 0 & 0 & 0 & 0 & \frac{1}{G_{\mathrm{xy}}} \end{array}\right)\left(\begin{array}{c} \sigma_{\mathrm{xx}}\\ \sigma_{\mathrm{yy}}\\ \sigma_{\mathrm{zz}}\\ \tau_{\mathrm{yz}}\\ \tau_{\mathrm{xz}}\\ \tau_{\mathrm{xy}} \end{array}\right)$$
where {ε} represents the vector of deformation components and {σ} contains the stress components. [S] corresponds to the compliance matrix, which includes the elastic modulus, Poisson’s ratio and shear modulus for the material main axis system.

Considering the printing principle of FDM AM materials, it is reasonable to simplify their mechanical behavior as transversely isotropic materials. The gradient between directions on the material layer plane is found to be relatively small in relation to the elastic constants for unreinforced thermoplastics [20]. The material layer is the transverse plane (XY) and the mechanical properties on this plane are the same while the mechanical properties in the direction that is perpendicular to the material layer (Z) are different from those on the transverse plane [21,22]. Transversely isotropic materials can be described by five independent elastic constants (E_T_, E_L_, ν_TT_, ν_TL_, G_L_) where T represents the transverse direction (material layer plane) and L represents the longitudinal or polar direction (perpendicular to the material layer plane). Comparing these elastic constants with the coefficients of the matrix in Equation (2), we obtain the following relationships.
(3)$$E_{T}=E_{x}=E_{y}$$
(4)$$E_{L}=E_{z}$$
(5)$$\nu_{\mathrm{TT}}=\nu_{\mathrm{xy}}=\nu_{\mathrm{yx}}$$
(6)$$\nu_{\mathrm{TL}}=\nu_{\mathrm{yz}}=\nu_{\mathrm{xz}}$$
(7)$$G_{\mathrm{xy}}=E_{T}/2\left(1+\nu_{T}\right)$$
(8)$$G_{\mathrm{LT}}=G_{\mathrm{yz}}=G_{\mathrm{xz}}$$

Mechanical tensile and shear tests are conducted using the UNE-EN-ISO-527-2 [Y] and ASTM D 3039-76 standards, respectively [23,24], to obtain the elastic coefficients and material strength. The aforementioned items are used to define the dimensions of the test parts, test speeds and calculation of mechanical properties. The elastic module, E_i_, are derived from the stress–strain diagram, as shown in Equation (9):(9)$$E_{i}=\frac{{\Delta \sigma }_{i}}{{\Delta \varepsilon }_{i}}=\frac{\sigma_{i}^{\prime }-\sigma_{i}}{\varepsilon_{i}^{\prime }-\varepsilon_{i}}$$
where σ_i_ is the stress measured for a unit strain value ɛ_i_ = 0.05% and σ′_i_ is the stress measured for a strain value ɛ′_i_ = 0.25%. The Poisson coefficients, ν_ij_, are obtained as follows:(10)$$\nu_{\mathrm{ij}}=-\frac{\varepsilon_{i}}{\varepsilon_{j}}$$
where ɛ_j_ represents the longitudinal elongation per unit length in the axis direction of the sample, and ɛi corresponds to the shortening of a length located on a perpendicular plane to the direction of the applied load. Once the values of these elastic constants are obtained, the tensor’s symmetry should be validated according to the following relationship:(11)$$\frac{\nu_{\mathrm{ij}}}{E_{j}}=\frac{\nu_{\mathrm{ji}}}{E_{i}}$$

Finally, the shear modulus, G_LT_, can be derived by testing an off-axis specimen with θ = 10° printing angle on either the XZ or YZ plane and applying the following equation
(12)$$G_{\mathrm{LT}}=\frac{{\left(\sin\theta \right)}^{2}{\left(\cos\theta \right)}^{2}}{\frac{1}{E_{\theta }}-\frac{{\left(\cos\theta \right)}^{4}}{E_{T}}-\frac{{\left(\sin\theta \right)}^{4}}{E_{L}}+\frac{2\nu_{\mathrm{TL}}{\left(\sin\theta \right)}^{2}{\left(\cos\theta \right)}^{2}}{E_{L}}}$$

### 3.2. Tensile Tests

The tensile test consists of the application of stress in the same direction as the longitudinal axis of the samples. This test can set a graph with the obtained stress–strain data. The elastic and plastic behavior of a material can be specified from this data, as well as quantifying its maximum or breaking-point strength [25,26]. In order to take into account the printing direction for the constitutive material behavior of the printed part, all manufacturing directions were carried out under the same conditions: 50% infill and rectilinear pattern with 45° orientation fillings. Figure 4 shows the different orientation of the printed parts according to the UNE-EN-ISO–ASTM 52921:2017 standards, where flat, edge and vertical building configurations are shown. 

It should be noted that the transverse elastic modulus, E_T_, may be determined along the tensile specimen’s longitudinal direction for a specimen manufactured lying in the building space (flat configuration) or for one rotated by 90° along the longitudinal axis (edge configuration). Figure 5 shows that two redundant configurations result in each main direction representing the transverse elastic modulus, E_T_. This research will employ the average of both values in order to derive a single transversally isotropic matrix that can be applied for the purposes of numerical calculation. As regards the three Poisson’s ratio obtained, whether the symmetry condition of the compliance tensor is substantiated with minimal deviation will be checked.

Figure 6 shows the vertical configuration of the tensile specimen for derivation of Ez, ν_TL_ and G_LT_. An off-axis test specimen in which the transverse and longitudinal directions do not correspond with the load direction is used to determine the shear modulus G_LT_, which is inclined by an angle of 10° to either X or Y axes.

Five samples were printed and tested according to UNE standard for laminated materials, as seen in Figure 7 and Figure 8. 

The test was performed using a universal machine at 2 mm/min measuring force and longitudinal and transverse deformation. A DIC (digital image correlation) optical method system was used to measure the deformation (Figure 9). 

Before conducting the tests, a random dot map was sprayed on the surface. This DIC system analyzes the movement of each of the dots and determines the level of deformation of each area [27]. Figure 10 shows the strain distribution on a sample under tensile load.

## 4. Discussion

### 4.1. Obtaining the Anisotropic Matrix

Samples printed in the three directions with a 50% rectilinear infill and 0.2 mm layer height were tested under the conditions shown in Table 3.

The values of the transverse elastic coefficients are obtained as the average of the ones obtained for each parameter and configuration (Table 4). Table 5 shows the final values of the elastic constants for the characterized material. The symmetry condition of Poisson’s ratios was validated according to Equation (11). The percentage deviation is 1%, which is acceptable and theoretically of little importance for further use in a finite element program.

Table 6 shows the tensile strength and the elastic constants of the orthotropic matrix which we used to feed the material data needed for ANSYS. Since the results are different in the three directions, an isotropic approximation in order to have single values for E, ν and G, would lead us to calculation errors. Concerning tensile strengths, the tension values in the flat and edge configurations are similar and significantly higher than the one obtained in the L configuration because the tensile test in L configuration is perpendicular to the building direction. Therefore, each layer is made of filaments laid on the perpendicular direction so the sample has lower resistance than in the other configuration where the filaments are laid in the same direction as the force applied in the tensile test. Although the final configuration is the same in both flat and edge configurations, the values are higher in the edge due to two factors: heat diffusion and contour layer.

The first one, heat diffusion, relates to speed and time. The print speed was the same in all configurations. Therefore, in the case of edge configuration, as the manufacturing section is smaller, the machine deposited one layer over another in less time. Thus, the heat diffusion in this configuration was greater and favored union between layers. Therefore, E_T_^(edge)^ and E_L_ are similar as the section is small in both configurations. In the flat configuration, which takes longer, the material cools down and that weakens the union. The second one is related to the contour layer; the more contour layers the stronger the configuration. 

The fracture produced was analyzed using SEM microscopy to check whether the piece was printed correctly and if the pattern and percentage of infill remained constant during printing. The selected print layer height is 0.2 mm and the infill percentage is 50%. Figure 11 shows that the printing kept the defined pattern constant. Figure 12 shows distance between the layers or layer height: 0.22 mm. Therefore, the internal dimensions were maintained, and the printing was validated. 

In order to validate the obtained constitutive matrix, a comparison was made between a finite element simulation of a flexural test according to UNE-EN-ISO 178 2011 and the flexural experimental test using five printed samples. 

The flexural test was performed according to standard specifications and test procedure, where the flexion resistance was determined as well as the stress/strain ratio. The test consists of applying a load on a rectangular section sample that rests on two supports and is flexed by a load element that acts on the center of the tested part. The part is flexed at a constant speed until it breaks or reaches the maximum deformation of 5%. The values measured during the test are the F and the resulting arrow indicating the direction executed on the center, as shown in Figure 13. 

A total of five samples measuring 80 mm × 10 mm × 4 mm were printed. Figure 14 shows the printing of the samples which were printed with the defined optimal printing parameters. 

The flexural test was carried out on the printed samples as shown in Figure 15. 

The results of the tests are shown on Table 7, where the deformation is from 1 to 1.03 mm for an average force of 18N. This value is similar for the five samples that were printed.

### 4.2. Validation by ANSYS Flexion Simulation

We needed to model the boundary conditions, i.e., the supports design and the applied load, with the aim to replicate the test conditions more accurately in order to implement a proper ANSYS validation [28]. The dimensions for the test samples were those required by the standards (80 × 10 × 4 mm). In order to represent the real conditions of the test, a support contact condition with a sliding allowance was considered between the samples and the supports, as well as between the indenter and the sample. The base of the supports was recessed and the load was applied to the indenter, which was only allowed to move in the vertical direction. The type of calculation was linear-elastic, since there were no nonlinearities, or large deformations, or displacements due to the material itself. The material was defined by the orthotropic matrix obtained in Table 5 and it followed the procedures set by the UNE-EN-ISO 178-2011 standard for flexion. The meshing of the model was performed using hexahedral elements as the geometry was simple and the precision per cost relation with a hexahedral mesh was properly balanced. For the discretization of the beam, 120 hexahedral elements were used. Table 8 summarizes the input for the Ansys numerical analysis and Figure 16 shows the simulation performed for the flexural test where the maximum displacement produced under a perpendicular load of 18 N is shown. The final result was 1.03 mm.

Figure 17 shows a comparison of the experimental testing with the ANSYS simulation for the linear region, to validate the force displacement diagram obtained experimentally and the one obtained numerically. Both diagrams are nearly superimposed, which validates the conclusions and the matrix obtained.

The characterized constitutive matrix was contrasted by performing a simulation with finite elements and a flexural test—UNE-EN-ISO 178-2011—on five printed samples. In the experiment previously performed, the resulting displacement was 1033 mm, which was the same as the result obtained in the ANSYS simulation. Therefore, we can conclude that the obtained elastic coefficient fits the ones obtained by experimental trials.

## 5. Conclusions

This article studied the mechanical properties of the thermoplastic polyphenylene polysulfide (PPS) printed using FDM filament deposition technology to establish a guide that will enable designers to select those parameters that best fit the mechanical properties sought in the printed part. The optimal printing parameters related to temperature, infill percentage, pattern, and layer height were studied. Once these parameters were set, a transversally isotropic matrix was proposed to approximate the orthotropic behavior of the PPS material manufactured by fuse deposition. This symmetry condition was also demonstrable, so only five, instead of nine independent characteristic values, were needed to describe the material behavior for this special case of material orthotropy. Finally, the approximated constitutive matrix of the material was defined. 

An analysis by SEM microscopy was performed to validate the correct impression inside the samples, checking that the rectilinear pattern, the infill percentage and the layer height remained constant during printing. The resulting constitutive matrix was validated by comparing the resulting values from the flexural test on five printed samples and the values obtained from the ANSYS simulation on the characterized material. It can be concluded that the values obtained for the elastic coefficients assuming transversally isotropic behavior are valid as the experimental results agree with the simulated ones. Although the elastic constants of the characterized material are not very far from the isotropic behavior, designers may use the values proposed in this work to obtain more accurate results when performing a numerical simulation. The tensile strength values obtained in both flat and edge configurations are similar but higher in the edge configuration due to two factors: heat diffusion and contour layer.

Our research did not analyze the influence of an implemented annealing treatment. It should, nevertheless, be noted that, in order to achieve maximum performance in printed PPS parts, these should undergo a heating treatment that increases crystallinity so they reach maximum mechanical, thermal and chemical resistance. Further research could be carried out in the future to obtain the orthotropic matrix with different materials, patterns and percentages. Subsequent studies could also focus on researching how pattern and infill percentage affect mechanical properties, and/or which the optimum printing speed on each direction is.

## Figures and Tables

**Figure 1 materials-14-01123-f001:**
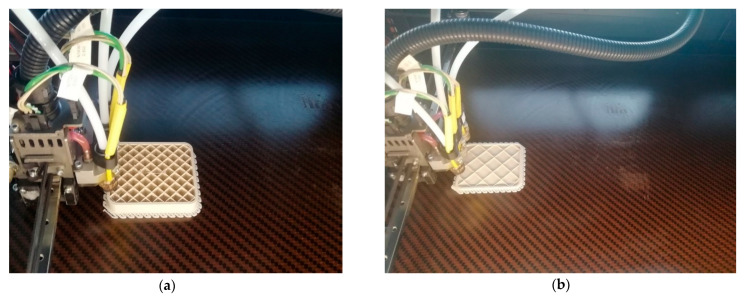
(**a**) Phenylene polysulfide thermoplastic (PPS) infill 10% rectilinear printing; (**b**) PPS infill 5% rectilinear printing.

**Figure 2 materials-14-01123-f002:**
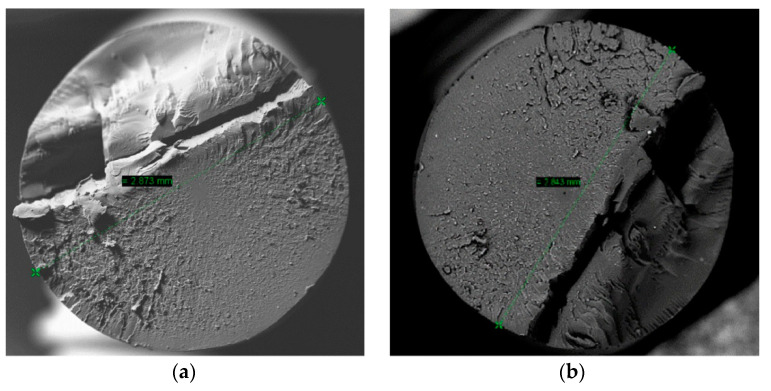
Filament sections in different zones. (**a**) Diagonal Measure 1 (**b**) Diagonal Measure 2.

**Figure 3 materials-14-01123-f003:**
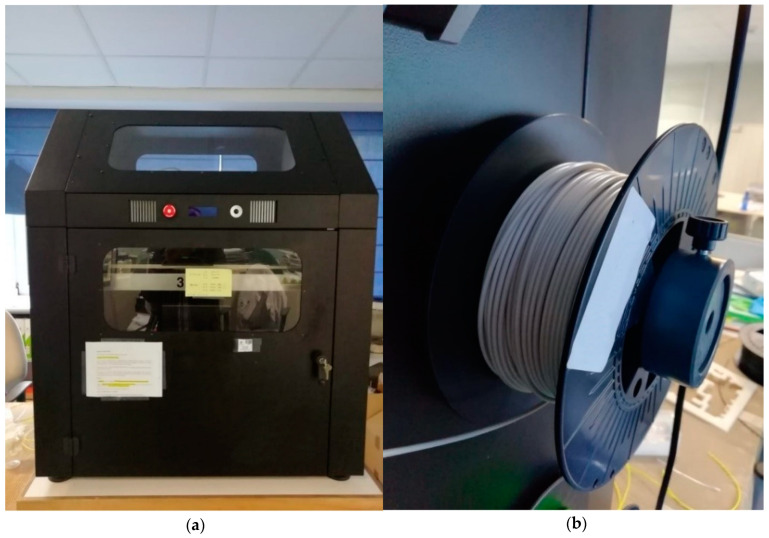
(**a**) 3NTR 3D printing machine; (**b**) PPS material.

**Figure 4 materials-14-01123-f004:**
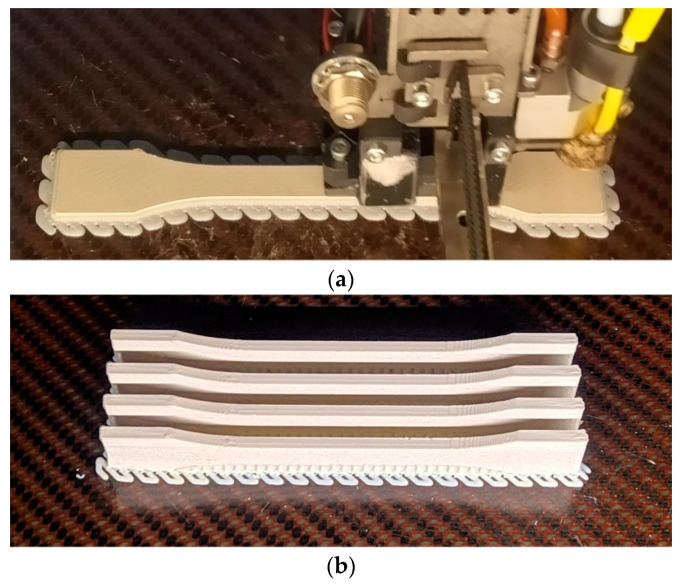
Manufacturing directions. (**a**) Flat configuration; (**b**) edge configuration; (**c**) vertical configuration; (**d**) 10° orientation.

**Figure 5 materials-14-01123-f005:**
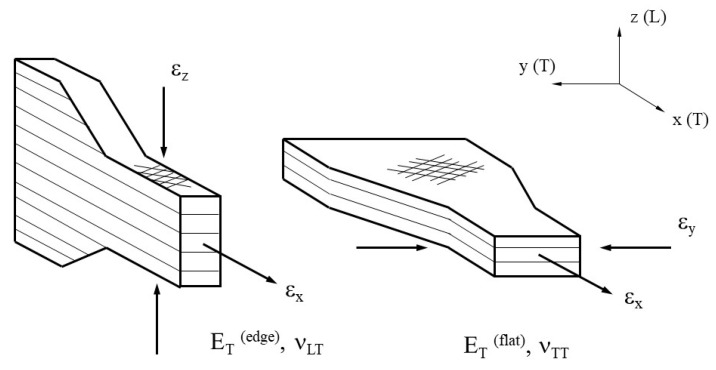
Flat and edge configurations of tensile specimen to determine E_T_^(edge)^, E_T_^(flat)^, ν_LT_, and ν_TT_.

**Figure 6 materials-14-01123-f006:**
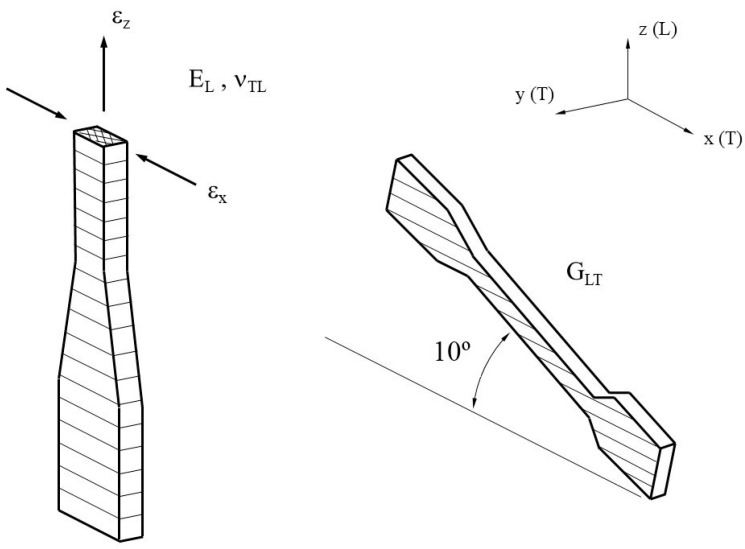
Configuration of tensile specimen on vertical and 10° configuration to determine Ez, ν_TL_, and G_LT._

**Figure 7 materials-14-01123-f007:**
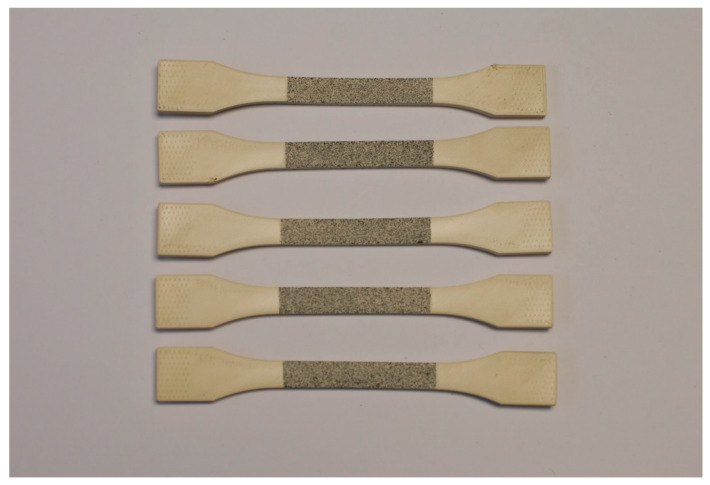
Printed samples for tensile test.

**Figure 8 materials-14-01123-f008:**
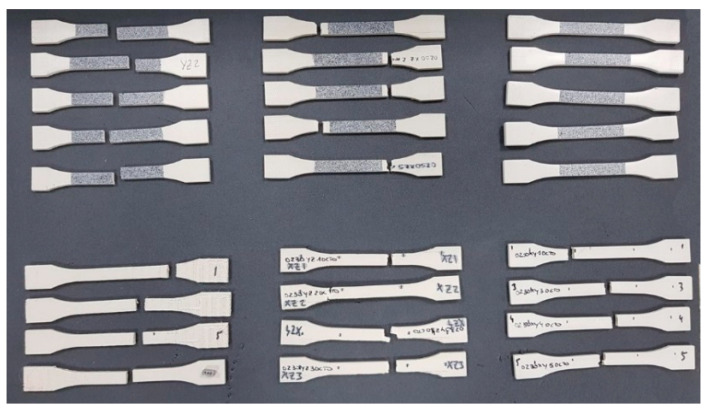
Tested samples after failure.

**Figure 9 materials-14-01123-f009:**
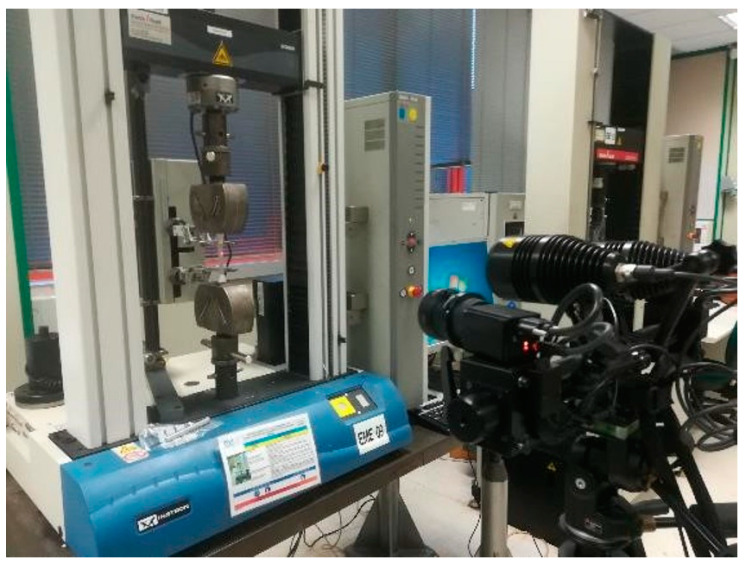
Digital image correlation (DIC) optical system.

**Figure 10 materials-14-01123-f010:**
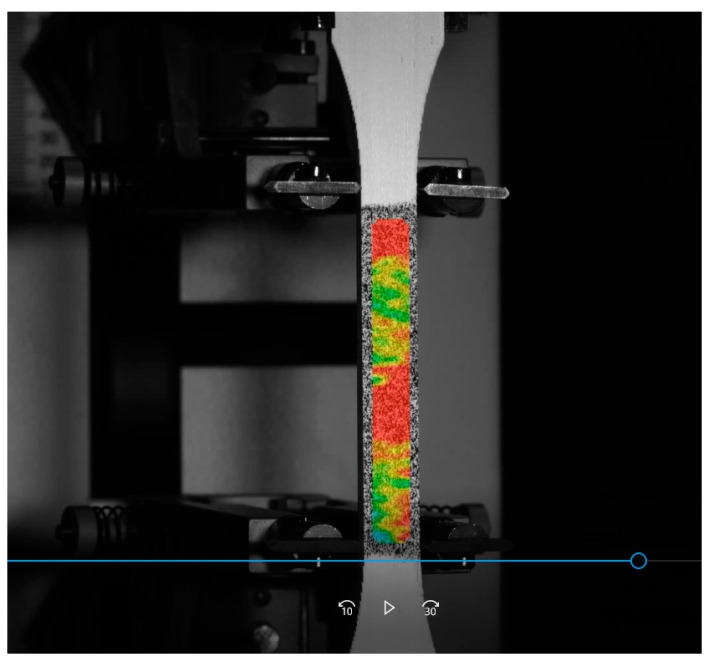
DIC strain distribution of a sample.

**Figure 11 materials-14-01123-f011:**
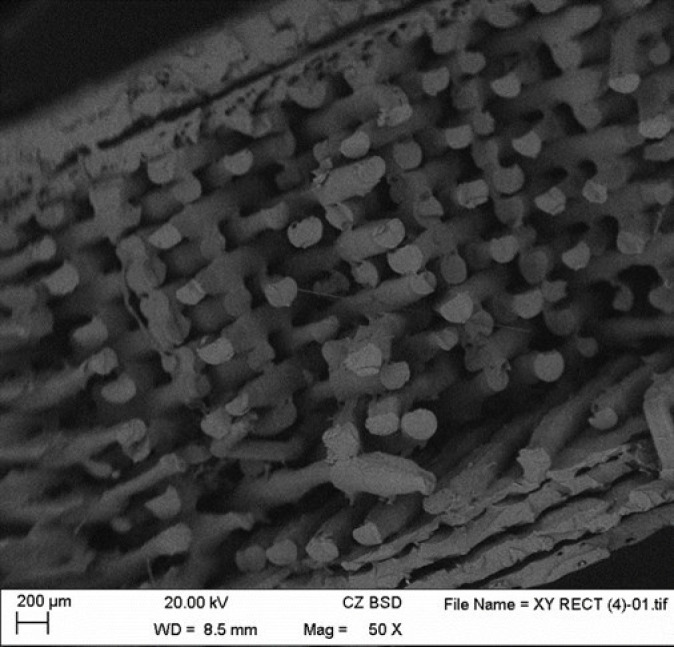
Rectilinear pattern.

**Figure 12 materials-14-01123-f012:**
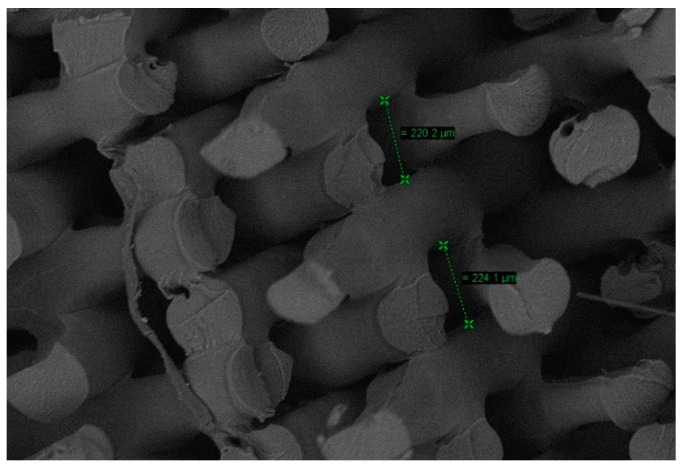
Distance between layers.

**Figure 13 materials-14-01123-f013:**
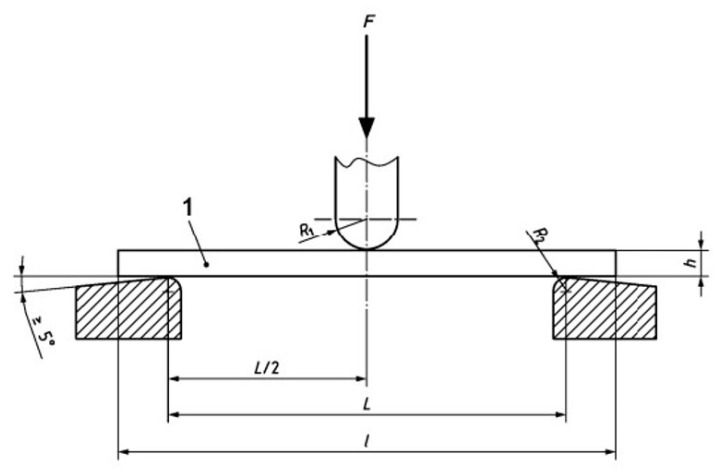
Flexural test according to the UNE-EN ISO 178-2011 normative.

**Figure 14 materials-14-01123-f014:**
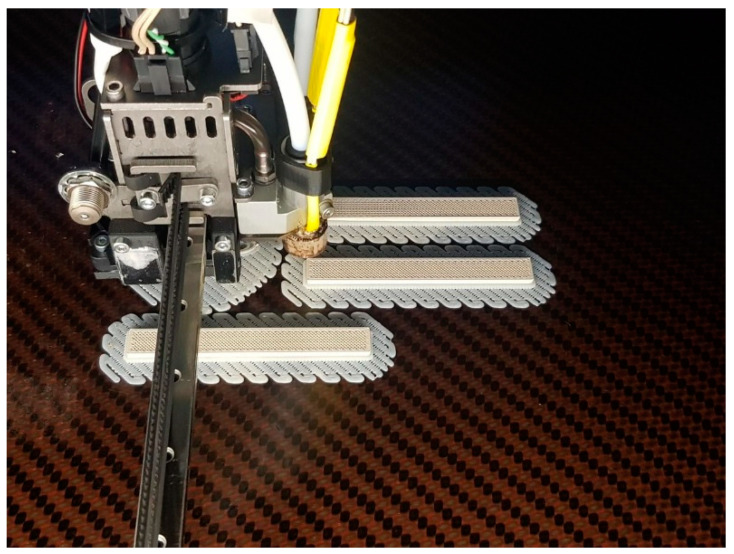
Printing of the flexural samples.

**Figure 15 materials-14-01123-f015:**
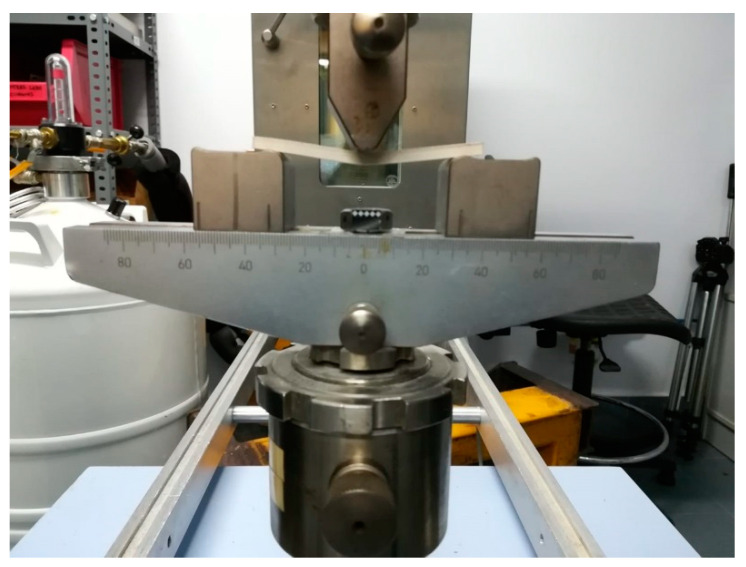
Flexural test according to UNE-EN-ISO 178-2011 standards.

**Figure 16 materials-14-01123-f016:**
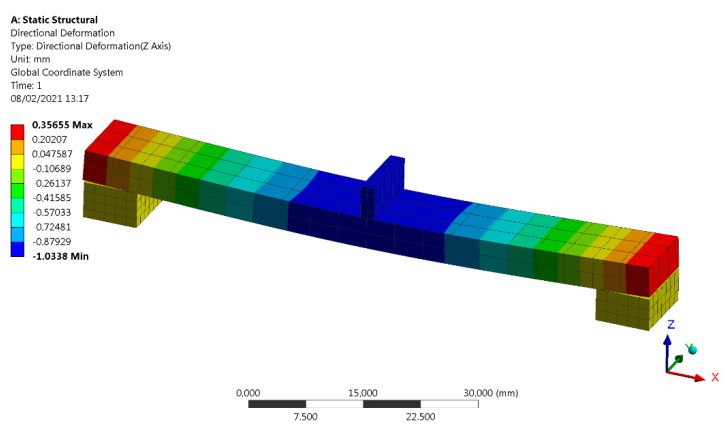
Finite elements simulation of flexural test part for a load of 18 N.

**Figure 17 materials-14-01123-f017:**
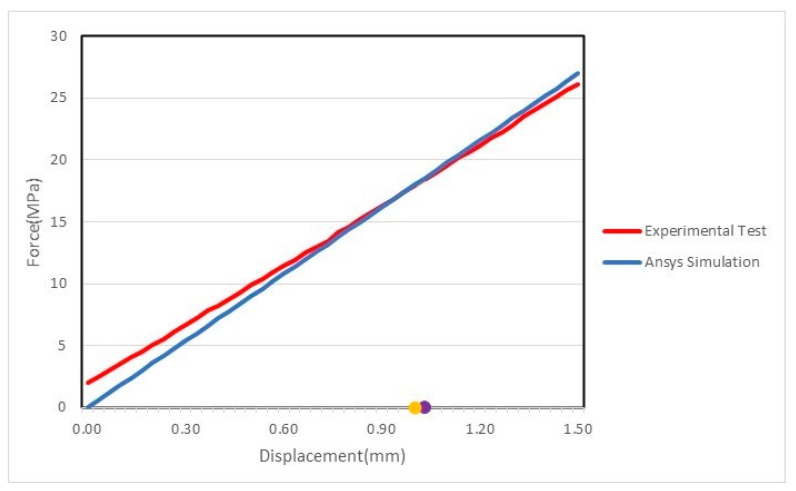
Experimental testing and ANSYS simulation comparison.

**Table 1 materials-14-01123-t001:** Mechanical properties of different commercial materials, Stratasys.

Material	Resistance to XZ Axis Traction	Resistance to ZX Axis Traction	Module to XZ Axis Tensile	Module to ZX Axis Tensile	Resistance to XZ Flex	Resistance to ZX Flex	Module to XZ Flex	Module to ZX Flex	Elongation to XZ Axis Breakage	Elongation to ZX Axis Breakage	Vitreous Transition Temperature
ULTEM 1010	64 MPa	42 MPa	2.770 MPa	2.200 MPa	144 MPa	77 MPa	2.820 MPa	2.230 MPa	3.3%	2%	215 °C
ULTEM 9085	47 MPa	33 MPa	2150 MPa	2270 MPa	112 MPa	68 MPa					
PPSF	55 MPa		2.100 MPa		110 MPa		2.300 MPa	250 MPa	5.8%	2.2%	186 °C
PC-ISO	57 MPa		2.000 MPa		90 MPa		2.200 MPa		4%		230 °C
PC-ABS	41 MPa		1.900 MPa		68 MPa		1.900 MPa		6%		161 °C
PC	40 MPa	30 Mpa	1.944 MPa	1958 MPa	89 MPa	68 MPa	2.006 MPa	1.800 MPa	2.2%	2%	125 °C
Nylon 12	32 MPa	26 Mpa	1.282 MPa	1138 MPa	67 MPa	61 MPa	1.276 MPa	1.180 MPa	2.4%	2.7%	161 °C
Nylon 6	49.3 MPa	28.9 Mpa	2.232 MPa	1817 MPa	97.2 MPa	82 MPa	2.196 MPa	1.879 MPa	2.3%	1.7%	
ASA	29 MPa	27 Mpa	2.010 MPa	1950 MPa	60 MPa	48 MPa	1.870 MPa	1.630 MPa	2%	2%	108 °C
ABS-M30i	36 Mpa		2.400 MPa			61 MPa	2.300 MPa		4%		108 °C
ABS-M30	31 MPa	26 Mpa	2.230 MPa	2180 MPa	60 MPa	48 MPa	2.060 MPa	1.760 MPa	7%	2%	108 °C
ABSi	37 MPa		1.920 MPa		62 MPa		1.920 MPa		4.4%		116 °C

**Table 2 materials-14-01123-t002:** PPS technical data sheet.

PPS Mechanical Properties			
Type	Test Method	Imperial	Metric
Tensile Modulus	ASTM D638	285144.2 psi	1966 MPa
Yield Point	ASTM D639	39.7%	39.7%
Tensile Elongation at Yield	ASTM D640	2.8%	2.8%
Tensile Strneght Ultimate	ASTM D641	5018.306 psi	34.6Mpa
Tensile Elongation at Break	ASTM D642	7.4%	7.4%

**Table 3 materials-14-01123-t003:** Testing conditions.

Parameter	Value
Room conditions	23 °C and 40% HR
Universal test machine	Zwick/Roell
Load shell	10 kN
Test speed	2 mm/min
Distance between jaws	119 mm
DIC system	GOM-ARAMIS
DIC volume analysis	125 × 100 mm
DIC distance	845 mm
DIC sampling frequency	1 Hz

**Table 4 materials-14-01123-t004:** Results obtained for each configuration.

Young’s Modulus	Value (Gpa)	Poisson’s Coefficient	Value	Shear Modulus	Value (Gpa)	Tensile Strength	Value (Mpa)
E_T_^(flat)^	1.38	ν_TT_	0.40	G_TT_^(flat)^	0.49	σ_T_^(flat)^	29.2
E_T_^(edge)^	1.58	ν_LT_	0.39	G_TT_^(edge)^	0.56	σ_T_^(edge)^	32.3
E_L_	1.60	ν_TL_	0.40	G_LT_	0.67	σ_L_	15.41

**Table 5 materials-14-01123-t005:** Final values.

Young’s Modulus	Value (Gpa)	Poisson’s Coefficient	Value	Shear Modulus	Value (Gpa)	Tensile Strength	Value (Mpa)
E_T_^(average)^	1.48	ν_TT_	0.4	G_TT_^(average)^	0.52	σ_T_^(average)^	30.8
E_L_	1.60	ν_LT_	0.39	G_LT_	0.67	σ_L_	15.41
		ν_TL_	0.40				

**Table 6 materials-14-01123-t006:** Material properties matrix.

Elastic Coefficient	Value
E_T_	1.48 GPa
E_L_	1.60 GPa
ν_TT_	0.40
ν_LT_	0.39
G_LT_	0.67 GPa

**Table 7 materials-14-01123-t007:** Displacement values for flexural test.

Sample 1			Sample 2		Sample 3		Sample 4		Sample 5	
Time	Force	Displacement	Force	Displacement	Force	Displacement	Force	Displacement	Force	Displacement
seg	N	mm	N	mm	N	mm	N	mm	N	mm
0	2.026	0.000	2.002	0.000	2.027	0.000	2.127	0.000	2.0011	0.0000
10	7.234	0.333	7.484	0.333	7.068	0.332	7.137	0.332	7.2551	0.3333
20	12.506	0.667	12.867	0.667	12.325	0.666	12.744	0.666	12.6871	0.6667
25	15.201	0.833	15.606	0.833	15.077	0.832	15.455	0.832	15.3430	0.8333
30	17.907	1.000	18.543	1.000	17.845	0.999	18.293	0.999	18.1246	1.0000
31	18.427	1.033	19.170	1.033	18.347	1.032	18.750	1.032	18.7516	1.0334
40	23.508	1.333	24.257	1.333	23.427	1.332	23.855	1.332	23.7171	1.3333
45	26.160	1.500	27.130	1.500	26.274	1.499	26.737	1.499	26.4891	1.5000

**Table 8 materials-14-01123-t008:** Input of the Ansys simulation.

Elastic Coefficient	Value
E_x_	1.48 GPa
E_y_	1.48 GPa
E_z_	1.6 GPa
ν_xy_	0.40
ν_yz_	0.40
ν_xz_	0.40
G_xy_	0.52 GPa
G_yz_	0.67 GPa
G_xz_	0.67 GPa

## Data Availability

Data sharing not applicable.

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
