# Peer review of "Identifying Elastic Constants for PPS Technical Material When Designing and Printing Parts Using FDM Technology"

_materials, 2021, doi:10.3390/ma14051123_

Round 1

Reviewer 1 Report

There are some weaknesses through the manuscript which need improvement. Therefore, the submitted manuscript cannot be accepted for publication in this form, but it has a chance of acceptance after a major revision. My comments and suggestions are as follows:

1- Abstract gives information on the main feature of the performed study, but some details about the experimental tests (at least in a couple of sentences) should be added. However, a concise abstract is needed.

2- It is not necessary to mention standards number in abstract. Moreover, it would be nice if authors use an appropriate tittle for the manuscript.

3- Authors must clarify necessity of the performed research. Aims and objectives of the study, must be clearly mentioned in concluding part of introduction.

4- The literature study must be enriched. It is highly recommended to read and cite the published papers: (a) https://doi.org/10.1002/nme.4936 and (b) https://doi.org/10.1016/j.eml.2020.100692 

5- It is necessary to add references for Table 1. Moreover, a column must be added to show name of the conducted tests.

6- Fig. 2 must be removed from introduction. As it shows printing process, can be illustrated in Section 2. Scale?

7- Name and logo of the company (nPOWER) must be removed from Fig. 3. Please make a table and write the mechanical properties of material (instead of a figure with logo of the company). Moreover, name of company in Fig. 4 (b) must not be shown. (current version is like an advertisement).

8- The main reference of each formula must be cited. All the parameters in each formula must be introduced.

9- As DIC system was used, its result must be illustrated. For example, a figure (specimen) with strain distribution under tensile load. (showing specimens prepared for DIC is not enough)

10- Scale in Fig. 11 and 12 must be added. Moreover, details of the simulation must be added (input of the ANSYS simulation in a table). Why this size of the mesh was selected.

11- In its language layer, the manuscript should be considered for English language editing. There are sentences which have to be rewritten.

12- The conclusion must be more than just a summary of the manuscript. Please provide all changes in text and reference update based on all recommended papers by red color in the revised version.

Reviewer 2 Report

The topic of the paper is interesting, but in my opinion it should not be published in its present form. Besides the English language that must be improved there is a serious problem with the application of the ISO 14129 norm for the determination of the shear characteristics. The norm clearly states that the specimen must be fabricated as a +45° -45° balanced and symmetric laminate. It is clear from Fig. 7 that only the +45° orientation has been considered. As a result the determination of the shear characteristics cannot be considered correct. It is my opinion that a balanced and symmetric laminate cannot be obtained by the authors with their setup, therefore i strongly recommend they use a 10° tension test (no symmetry needed) to evaluate shear properties.

As a further remark, the elastic properties do not appear very anisotropic from Tab. 5, the authors should not state that the properties are "very different". On the other hand, strength properties appear indeed very different. A discussion about the different behavior between elastic properties and strength properties would be beneficial.

Last remark. In the FEM validation using ANSYS, the number of elements (or the number of nodes) should be specified, and the authors should compare the experimental testing with the simulation through a figure showing the force displacement diagram obtained experimentally and the one obtained numerically. Comparing just at one point, i.e. 18N, is not acceptable.

In conclusion, the paper in my opinion is not acceptable in this form, but the authors have the possibility to improve it substantially if they follow my suggestions. 

Round 2

Reviewer 1 Report

Please note that format to reference journal articles in the reference list must be as follows: (you should correct all references)

Last name, First name. Article title. Journal Name, Volume, Page range, DOI.

The paper has been improved and corresponding modifications have been conducted. In my opinion, the current version can be considered for publication in Materials.

Reviewer 2 Report

The authors have performed many of the corrections i have asked. I think the most important improvement has been incorporated into the new manuscript, therefore i am not against publication of the article as it is, even if i still have some doubts, for example on the longitudinal modulus being higher than the two transverse moduli.